# Terminal Sliding Mode Control with a Novel Reaching Law and Sliding Mode Disturbance Observer for Inertial Stabilization Imaging Sensor

**DOI:** 10.3390/s20113107

**Published:** 2020-05-31

**Authors:** Xin Che, Dapeng Tian, Ping Jia, Yang Gao, Yan Ren

**Affiliations:** 1Key Laboratory of Airborne Optical Imaging and Measurement, Changchun Institute of Optics, Fine Mechanics and Physics, Chinese Academy of Sciences, Changchun 130033, China; chexin15@mails.ucas.edu.cn (X.C.); jiap@ciomp.ac.cn (P.J.); gaoyang161@mails.ucas.edu.cn (Y.G.); 2University of Chinese Academy of Sciences, Beijing 100049, China; 3Information Engineering School, Inner Mongolia University of Science and Technology, Baotou 014010, China; ren0831@imust.edu.cn

**Keywords:** terminal sliding mode control (TSMC), reaching law method, high-order terminal sliding mode observer (HOTSMO), disturbance rejection control, inertial stabilization imaging sensor (ISIS)

## Abstract

High-performance control of inertial stabilization imaging sensors (ISISs) is always challenging because of the complex nonlinearities induced by friction, mass imbalance, and external disturbances. To overcome this problem, a terminal sliding mode controller (TSMC) based on a novel exponential reaching law (NERL) method with a high-order terminal sliding mode observer (HOTSMO) is suggested. First, the TSMC based on NERL is adopted to improve system performance. The NERL incorporates the power term and switching gain term of the system state variables into the conventional exponential reaching law, and the convergent speed of the TSMC is accelerated. Then, an HOTSMO is designed, which considers the speed and lumped disturbances of the system as the observation object. The estimated disturbance is then provided as a compensation for the controller, which enhances the disturbance rejection ability of the system. Comparative simulation and experimental results show that the proposed method achieves the best tracking performance and the strongest robustness than PID and the traditional TSMC methods.

## 1. Introduction

An inertial stabilization imaging sensor (ISIS) is widely applied in robot systems to achieve stable image acquisition under mobile platform, such as wheeled robot, aerial robot, simulation robot, etc. [1,2]. The ISIS isolates nonideal angle disturbances and maintains the line of sight (LOS) of stabilized optical sensors [3]. It is generally affected by disturbance factors, including internal disturbances caused by mass imbalance torque [4], friction torque [5], and cable restraint torque [6], and by external disturbances caused by carrier motion or vibration. Moreover, when environment change occurs, system parameters vary. Therefore, the ISIS system is a nonlinear time-varying system with parameter perturbation and multi-source complex disturbances. It is difficult to achieve satisfactory performance in the entire operating range only using traditional linear PID control schemes. Thus, an effective control method is considered to be of great practical significance for improving the dynamic response and disturbance rejection performance of the ISIS system.

In recent years, owing to the rapid progress in digital signal processors, various modern control methods, such as robust control [7], active disturbance rejection control [8], adaptive control [9], back-stepping control [10], sliding mode control (SMC) [11], and intelligent control [12], have been proposed to improve the performance of the ISIS system. Scholars employed inexactly scheduling method to deal with parameter uncertainties [13,14]; however, it is difficult to address the lumped disturbance considering only parameter uncertainties. Therefore, SMC is chosen as a candidate for the controller. It is proven to be effective for maintaining system stability and consistent performance in the presence of parameter perturbation and disturbances [15]. However, the convergence rate of linear sliding surface is exponential with an infinite settling time. Thus, a terminal sliding mode control (TSMC) method was proposed with a nonlinear sliding surface that ensures that the states converge to the origin in finite time [16]. The finite-time convergence demonstrates that the TSMC has the advantage of a fast dynamic response and better robustness properties. In addition, a nonsingular TSMC [17] scheme was developed to avoid the singularity problem of the TSMC.

The traditional SMC usually adopts a constant reaching law with fixed switching gain; therefore, there is an upper limit to the disturbance rejection ability. Since the reaching law approach is directly related to the reaching process, a reasonable design of the reaching law can efficiently improve system performance [18]. Hence, many scholars have attempted to modify the reaching law by making discontinuous switching gain a function of the sliding mode variable to enhance the anti-disturbance ability. In [19], a novel reaching law was proposed in the robot system, which allows chattering reduction on the control input while maintaining a high tracking performance of the controller in a steady-state regime. In [20], an improved exponential reaching law was adopted in the permanent magnet synchronous motor control system, and it achieved higher robustness with lower chattering. A sliding-mode control strategy using a new reaching law was proposed in [21]. The new reaching law effectively suppressed sliding-mode chattering and increased the convergence rate of the system state reaching the sliding-mode surface. The performance of permanent magnet synchronous motor was improved by using their proposed method. However, in the aforementioned reaching law, the switching gain decreased when approaching the sliding surface; thus, the robustness of the controller near the sliding surface was reduced, and the reaching time increased. There is a tradeoff between chattering and disturbance rejection, and the method will reduce the robustness of the system, especially when there are multi-source complex disturbances in the ISIS system.

To address the aforementioned problem, the TSMC based on a novel exponential reaching law (NERL), which adjusts the controller gain based on the error signal between the actual and desired system states, is proposed in this paper. When the states of the system are near or far from the sliding surface, the switching gain increases, forcing the system states to move to the desired states rapidly. The convergent speed of the controller is increased, and the disturbance rejection ability is elevated. Moreover, to further improve the disturbance rejection performance and avoid the excessive switching gain of the TSMC, a high-order terminal sliding-mode observer (HOTSMO) is also designed. The estimated disturbance is given as the compensation part of the controller; the controller avoids the selection of excessive switching gain, thus avoiding system chattering caused by the TSMC. A composite controller combining the TSMC part based on the NERL and a compensation part based on the HOTSMO is developed. Simulation and experimental results verify the effectiveness of the proposed method.

The remainder of this paper is organized as follows. Section 2 describes the mathematical model of the ISIS. Section 3 provides the design process of the NERL-based TSMC and HOTSMO. Section 4 presents the simulations implemented to verify the effectiveness of the proposed method. Section 5 presents the experimental results. Finally, Section 6 concludes the study.

## 2. Mathematical Model of ISIS

The ISIS system is typically composed of two or three degrees of freedom, which can be decoupled structurally. The motion of each degree of freedom is considered separately to simplify the discussion. This study considers the single pitch axis as an example, and the schematic of the ISIS system is shown in Figure 1. The pitch axis is driven by a torque motor; the motor stator of ISIS is embedded in the carrier, which indicates that the stator and carrier are fixed as a rigid body. Moreover, the optical encoder is mounted on the axis to provide relative angular displacements of gimbals, and a gyro is applied to provide the angular velocity information of the axis. Based on the information measured by the gyro and encoder, the controller generates the corresponding control signals to adjust the motion of the pitch axis. Therefore, the LOS of the optical sensors is adjusted to obtain precise images and videos of targets. The dynamic model of an ISIS can be summarized as [22].
(1)$$R_{a}i+L_{a}\frac{di}{dt}+K_{e}\omega_{m}=u_{a}$$
(2)$$T_{M}=K_{t}i$$
(3)$$J_{m}{\dot{\omega }}_{m}+B_{m}\omega_{m}=T_{M}+T_{din}+T_{dex}$$
where $\omega_{m}$ denotes the angular speed of the motor, $u_{a}$ denotes the armature voltage, $R_{a}$ denotes the motor armature resistance, $L_{a}$ denotes the motor armature resistance, *i* denotes the armature current, $K_{e}$ denotes the coefficient of back-EMF, $K_{t}$ denotes the electromagnetic torque constant, $J_{m}$ denotes the moment of inertia, $B_{m}$ denotes the damping ration, and $T_{M}$ denotes the motor output torque. Further, $T_{din}$ denotes internal disturbances, including mass imbalance torque, friction torque, and sensors measurement error. $M_{dex}$ denotes the external turbulence, including carrier motion and vibration. It is reasonable to assume that $L_{a}\approx 0$ because the armature inductance is sufficiently small to be neglected.

According to Equations (Equation 1) and (Equation 2), we obtain,
(4)$$T_{M}=\frac{K_{t}}{R_{a}}\left(u_{a}-K_{e}\omega_{m}-\frac{L_{a}}{K_{t}}\frac{dT_{M}}{dt}\right)$$

Thus, Equation (Equation 3) can be rewritten as
(5)$$J_{m}{\dot{\omega }}_{m}+B_{m}\omega_{m}=\frac{K_{t}}{R_{a}}\left(u_{a}-K_{e}\omega_{m}\right)+T_{din}+T_{dex}$$

Setting $J=R_{a}J_{m}/K_{t},B=R_{a}B_{m}/K_{t}+K_{e},u=u_{a},T_{D}=R_{a}\left(T_{din}+T_{dex}\right)/K_{t}$, the dynamic equation can be presented as
(6)$$J{\dot{\omega }}_{m}+B\omega_{m}=u+T_{D}$$
where *J* is the equivalent moment of inertia, *B* is the equivalent damping ration, *u* is the equivalent input, and $T_{D}$ represents the disturbances including both internal disturbances and external disturbances.

If parameter uncertainties are considered, the dynamic equation of the ISIS can be rewritten as
(7)$$\begin{array}{c} J_{n}{\dot{\omega }}_{m}+B_{n}\omega_{m}=u+d \end{array}$$

Set parameter uncertainties $\Delta J=J-J_{n}$, $\Delta B=B-B_{n}$, where $J_{n}$ and $B_{n}$ are the moment of inertia and the frictional coefficient of the nominal model. *d* in Equation (Equation 7) can be expressed as $d=T_{D}-\Delta J{\dot{\omega }}_{m}-\Delta B\omega_{m}$, where *d* represents the lumped disturbances of the system including internal disturbances, external disturbances, and parameter uncertainties.

**Assumption** **1.***The derivative of d in Equation* (Equation 7) *with respect to time t can be regarded as bounded $\left|r\right|\leq l_{d}$, where $\dot{d}=r$ is the derivative of the lumped disturbances and $l_{d}$ is a positive constant [23]. It is reasonable since in a practical ISIS system, the system disturbances are considered to vary slowly compared with the system state in every sampling period of the speed loop.*

## 3. Composite Controller Design

In this section, a composite control method combining a TSMC with NERL and a compensation part based on HOTSMO is designed for the ISIS system.

The composite controller can be designed as
(8)$$u=u_{eq}+u_{n}-\hat{d}$$
where $u_{eq}$ denotes equivalent input, $u_{n}$ denotes switching input, and $\hat{d}$ denotes the estimated value of the lumped disturbances. The specific part will be introduced in the next section.

### 3.1. Terminal Sliding Mode Controller Design Based on Novel Exponential Reaching Law

The speed tracking error is defined as
(9)$$e=\omega_{r}-\omega_{m}$$
where $\omega_{r}$ is the reference speed.

The TSMC manifold is designed as
(10)$$s=e+\alpha {\dot{e}}^{p/q}$$
where α>0,p,q are positive odd integers, 1<p/q<2.

For system (7), the conventional exponential reaching law is adopted as
(11)$$\dot{s}=-k_{1}\mathrm{sign}\left(s\right)-k_{2}s$$
where $k_{1},k_{2}>0$ are constants. Therefore, the convergent characteristics of the system cannot achieve the best performance as the system states changes. Based on the system state *e* and manifold of the TSMC *s*, NERL is given by
(12)$$\dot{s}=-\frac{k_{1}}{N(s)}\mathrm{sign}\left(s\right)-k_{2}s$$

N(s) is designed as
(13)$$N\left(s\right)=\delta +\left(1-\delta \right)e^{-\mu |s|}$$
where 0<δ<1 and μ>0. Note that for any s∈R, 0<N(s)<1 is always satisfied. In this novel reaching law, it can be seen that if |s| increases, N(s) approaches δ; therefore, $k_{1}/N\left(s\right)$ converges to $k_{1}/\delta$, which is greater than $k_{1}$. This means that when the system state is away from the sliding surface, the attraction of the sliding surface will be faster than that in Equation (Equation 11), and the reaching time can be significantly shortened. However, if |s| decreases, N(s) approaches one, and $k_{1}/N\left(s\right)$ converges to $k_{1}$. Thus, when the system state approaches the sliding surface, $k_{1}/N\left(s\right)$ gradually decreases to weaken the chattering. Therefore, the improved exponential reaching law allows the controller to dynamically adapt to the variations in the switching function by letting $k_{1}/N\left(s\right)$ vary between $k_{1}$ and $k_{1}/\delta$.

Considering the speed error state Equation (Equation 9), the sliding surface Equation (Equation 10), and reaching law Equation (Equation 12), the TSM controller based on NERL can be designed as
(14)$$u_{eq}=J_{n}\left({\dot{\omega }}_{r}+\frac{B_{n}}{J_{n}}\omega_{m}\right)$$
(15)$$u_{n}=J_{n}\int_{0}^{t}\left[\frac{1}{\alpha }\frac{q}{p}{\dot{e}}^{2-p/q}+\frac{k_{1}}{N(s)}\mathrm{sign}\left(s\right)+k_{2}s\right]dt$$
(16)$$u=u_{eq}+u_{n}$$
The stability of the control system can be achieved by Theorem 1.

**Theorem** **1.***If Assumption 1 holds, under the control laws in Equations* (Equation 14)–(Equation 16)*, the speed tracking error of the ISIS system converges to zero in finite time, if the switching gain satisfies $k_{1}>l_{d}/J_{n}$.*


**Proof** **of** **Theorem** **1.**Choosing Laypunov function $V=s^{2}/2$, and differentiating *V* gives
(17)$$\begin{array}{cc} \dot{V} & =s\dot{s}=s\left(\dot{e}+\alpha \frac{p}{q}{\dot{e}}^{p/q-1}\ddot{e}\right)\\ \; & =s\alpha \frac{p}{q}{\dot{e}}^{p/q-1}\left(\ddot{e}+\frac{1}{\alpha }\frac{q}{p}{\dot{e}}^{2-p/q}\right)\\ \; & =s\alpha \frac{p}{q}{\dot{e}}^{p/q-1}\left({\ddot{\omega }}_{r}+\frac{B_{n}}{J_{n}}{\dot{\omega }}_{m}-\frac{\dot{u}}{J_{n}}-\frac{\dot{d}}{J_{n}}+\frac{1}{\alpha }\frac{q}{p}{\dot{e}}^{2-p/q}\right)\\ \; & =s\alpha \frac{p}{q}{\dot{e}}^{p/q-1}\left(-\frac{\dot{d}}{J_{n}}-\frac{k_{1}}{N(s)}\mathrm{sign}\left(s\right)-k_{2}s\right)\\ \; & \leq -\alpha \frac{p}{q}{\dot{e}}^{p/q-1}(\frac{|\dot{d}|}{J_{n}}\left|s\right|-k_{1}\left|s\right|+k_{2}s^{2})\\ \; & <-\alpha \frac{p}{q}{\dot{e}}^{p/q-1}(\left(k_{1}-\frac{l_{d}}{J_{n}}\right)\left|s\right|+k_{2}s^{2}) \end{array}$$Hence, for ${\dot{e}}^{p/q-1}\neq 0$, we can derive $\dot{V}<0$. Similar to the proof in [16], for ${\dot{e}}^{p/q-1}=0$, s≠0, from Equation (Equation 12), we obtain e≠0, which implies that ${\dot{e}}^{p/q-1}$ is not an attractor. Therefore, the state reaches the terminal sliding manifold s=0 from any initial condition in finite time.Supposing $t_{r}$ is the time when *s* reaches zero from s(0)≠0, once the sliding surface s=0 is reached, we have
(18)$$s=e+\alpha {\dot{e}}^{p/q}=0$$After a simple calculation, the time from s(0)≠0 to $e(t_{s})=0$ can be given as
(19)$$t_{s}=t_{r}+\frac{p}{\alpha^{-q/p}\left(p-q\right)}{\left|e\left(t_{r}\right)\right|}^{1-q/p}$$Therefore, the speed error can converge to zero in finite time $t_{s}$, which completes the proof. □

**Remark** **1.***Equation* (Equation 10) *implies that α determines the decay rate of the tracking error on the sliding surface, and it roughly determines the bandwidth of the tracking bandwidth, thus providing a faster response speed and higher tracking accuracy [24]. A large bandwidth will also amplify high-frequency noise. Similarly, a larger value of p/q results in a smaller convergence time as seen in Equation* (Equation 19)*; however, this will amplify the velocity measurement noises. The choice of $k_{1},\delta$ and μ requires a tradeoff between system robustness and chattering. $k_{2}$ increases the stiffness of the closed-loop system, and a large $k_{2}$ injects excess noise into the system.*


### 3.2. High-Order Terminal Sliding Mode Observer Design

The ISIS system is affected by multi-source complex disturbances. It is difficult to model and identify the lumped disturbances accurately; therefore, HOTSMO is implemented to improve the robustness of the ISIS system.

Regarding *d* as the system extended state, the mathematical model of ISIS system can be rewritten as
(20)$$\left\{\begin{array}{cc} {\dot{\omega }}_{m} & =\frac{1}{J_{n}}u-\frac{B_{n}}{J_{n}}\omega_{m}+\frac{d}{J_{n}}\\ \dot{d} & =r \end{array}\right.$$

Then, HOTSMO is designed as
(21)$$\left\{\begin{array}{cc} {\dot{\hat{\omega }}}_{m} & =\frac{1}{J_{n}}u-\frac{B_{n}}{J_{n}}{\hat{\omega }}_{m}+\frac{\hat{d}}{J_{n}}+u_{tsmo1}\\ \dot{\hat{d}} & =u_{tsmo2} \end{array}\right.$$
where ${\hat{\omega }}_{m}$ represents the estimated value of the velocity, $u_{tsmo1}$ represents the designed control law, and $u_{tsmo2}$ represents the designed equivalent lumped disturbances derivative. The speed estimated error ${\tilde{\omega }}_{m}$ and system disturbance estimated error $\tilde{d}$ are defined as
(22)$$\left\{\begin{array}{cc} {\tilde{\omega }}_{m} & =\omega_{m}-{\hat{\omega }}_{m}\\ \tilde{d} & =d-\hat{d} \end{array}\right.$$

By subtracting Equation (Equation 21) from Equation (Equation 20), we can obtain the equation for the derivative of the observation error as
(23)$$\left\{\begin{array}{cc} {\dot{\tilde{\omega }}}_{m} & =-\frac{B_{n}}{J_{n}}{\tilde{\omega }}_{m}+\frac{\tilde{d}}{J_{n}}-u_{tsmo1}\\ \dot{\tilde{d}} & =r-u_{tsmo2} \end{array}\right.$$

A terminal sliding surface is designed to achieve better tracking accuracy.
(24)$$s_{\omega }=\dot{\tilde{\omega }}+\beta {\tilde{\omega }}^{n/m}$$
where $s_{\omega }$ is the sliding surface, β>0, and m,n(m>n) are positive odd integers.

Thus, the HOTSMO control law can be obtained as
(25)$$\left\{\begin{array}{cc} u_{tsmo1} & =-\frac{B_{n}}{J_{n}}{\tilde{\omega }}_{m}+\beta {\tilde{\omega }}^{n/m}+v\\ \dot{v}+T_{w}v & =l_{1}\mathrm{sign}\left(s_{\omega }\right)\\ u_{tsmo2} & =l_{2}\mathrm{sign}\left(s_{\omega }\right) \end{array}\right.$$
where $s_{\omega }\left(0\right)=0$, $l_{1}$ is the control gain, $l_{2}$ is the feedback gain, and $T_{\omega }>0$ is the designed parameter. The block diagram of HOTSMO is given in Figure 2. The observer can obtain satisfactory observation performance by appropriately adjusting the parameters $l_{1}$, $l_{2}$, and $T_{\omega }$.

**Theorem** **2.***If Assumption 1 holds, for error system in Equation* (Equation 23)*, under the control law in Equation* (Equation 25)*, the observation error ${\tilde{\omega }}_{m}$ can converge to zero in finite time.*

**Proof** **of** **Theorem** **2.**The sliding-mode surface $s_{\omega }$ can be rewritten by substituting the first equation in Equation (Equation 23) and the first equation in Equation (Equation 25) into Equation (Equation 24) as
(26)$$s_{\omega }=\frac{\tilde{d}}{J_{n}}-v$$The derivative of $s_{\omega }$ can be calculated by combining the second equation in Equation (Equation 23) and the last two equations in Equation (Equation 25) as
(27)$$\begin{array}{cc} {\dot{s}}_{\omega } & =\frac{\dot{\tilde{d}}}{J_{n}}-\dot{v}\\ \; & =\frac{r-l_{2}\mathrm{sign}\left(s_{\omega }\right)}{J_{n}}-l_{1}\mathrm{sign}\left(s_{\omega }\right)+T_{\omega }v \end{array}$$By choosing the Lyapunov function $V_{0}=\frac{1}{2}s_{\omega }^{2}$, the derivative of $V_{0}$ is given as
(28)$$\begin{array}{cc} {\dot{V}}_{0} & =s_{\omega }{\dot{s}}_{\omega }=s_{\omega }\left(\frac{r-l_{2}\mathrm{sign}\left(s_{\omega }\right)}{J_{n}}-l_{1}\mathrm{sign}\left(s_{\omega }\right)+T_{\omega }v\right)\\ \; & =\left(\frac{rs_{\omega }-l_{2}\left|s_{\omega }\right|}{J_{n}}\right)-(l_{1}|s_{\omega }\left|+T_{\omega }vs_{\omega }\right) \end{array}$$By selecting $l_{1}>\left|T_{\omega }v\right|,l_{2}>l_{d}$, we can obtain
(29)$${\dot{V}}_{0}=s_{\omega }{\dot{s}}_{\omega }<0$$Therefore, the states of the system can converge to the TSM surface in finite time. When $s_{\omega }=0$, we have $\dot{\tilde{\omega }}+\beta {\tilde{\omega }}^{n/m}=0$. After simple calculation, the time from $s_{\omega }=0$ to reach ${\tilde{\omega }}_{m}=0$ can be given as
(30)$$t_{s}^{\prime }=\frac{m}{\beta (m-n)}{\left|\tilde{\omega }\left(0\right)\right|}^{(m-n)/m}$$This completes the proof. □

**Remark** **2.***Chattering Suppression Analysis: From the second equation of Equation* (Equation 21)*,*
(31)$$\hat{d}=\int u_{tsmo2}dt=\int l_{2}\mathrm{sign}\left(s_{\omega }\right)dt$$*Thus, the proposed observer smooths the estimated disturbance $\hat{d}$ by integrating the switching function. Simultaneously, from the first two equations in Equation* (Equation 25)*, it can be concluded that the chattering signal $l_{1}\mathrm{sign}\left(s_{\omega }\right)$ is smoothed by an equivalent low-pass filter with the bandwidth of $T_{\omega }$. Therefore, the proposed HOTSMO possess a smooth disturbance observation.*

## 4. Simulations

Simulations were implemented to verify the effectiveness of the proposed method.

### 4.1. Analysis of the Disturbances

The ISIS system is frequently subjected to various disturbances, which degrade its performance. Two types of major internal disturbances, mass unbalance torque and friction torque, were considered in the simulation.

#### 4.1.1. Mass Unbalance Torque

In a real ISIS system, the center of gravity $O_{2}$ will deviate from the rotating gimbal $O_{1}$, and there is an offset *r* caused by the lever arm between the center of gravity and the rotating gimbal, as shown in Figure 3. Thus, when the ISIS produces the deflecting angle $\theta_{p}$, the change in the deflecting angle will change the unbalanced force arm in the horizontal and vertical directions, forming the mass unbalanced torque.

The dynamic mass imbalance torque $T_{m}$ can be expressed as [25]
(32)$$T_{m}=ma_{y}r\sin\left(\theta \right)+m\left(g+a_{z}\right)r\cos\left(\theta \right)$$
where *r* represents the offset between the center of gravity and the rotating gimbal, $\theta_{p}$ represents the base angle related to the reference axis, *m* represents the mass of the whole system, $a_{y}$ and $a_{z}$ represent the horizontal acceleration and vertical acceleration, respectively.

#### 4.1.2. Friction Torque

Friction plays a major role in the ISIS system, and it limits the precision of the dynamic response of the system. The Stribeck model is adopted to describe the static and dynamic friction of the ISIS system. It is given as [26]
(33)$$T_{f}=\left[T_{c}+\left(T_{s}-T_{c}\right)e^{-{\left(\frac{\omega }{\omega_{s}}\right)}^{2}}\right]\mathrm{sign}\left(\omega \right)+\sigma \omega$$
where $T_{c}$ represents the coulomb friction torque, $T_{s}$ represents the maximum static friction, ω represents the relative angular velocity between contact surfaces, σ represents viscous friction coefficient, and $\omega_{s}$ represents the Stribeck velocity.

Hence, the designed controller requires strong robustness to deal with multi-source complex disturbances and maintain the dynamic response performance of the ISIS system.

The parameters of the ISIS system including parameter perturbation, mass imbalance, and friction torque are listed in Table 1.

Simulations were performed to evaluate the effectiveness of the proposed scheme, and the results are compared with those of the traditional PID controller and the TSM controller with traditional exponential reaching law under various operating conditions. The traditional TSMC can be designed as
(34)$$u_{n}^{\prime }=J_{n}\int_{0}^{t}\left[\frac{1}{\alpha }\frac{q}{p}{\dot{e}}^{2-p/q}+k_{1}\mathrm{sign}\left(s\right)+k_{2}s\right]dt$$
(35)$$u^{\prime }=u_{eq}+u_{n}^{\prime }$$

The parameters of the TSMC control methods and proposed method are listed in Table 2.

### 4.2. Case I—Sinusoidal Signal Tracking

For this case, three tests are conducted. The reference angular speed is set as sinusoidal signals with an amplitude of 10 °/s and a frequency of 1 Hz; 30 °/s and 1 Hz; and 30 °/s and 5 Hz. The experimental response curves are shown in Figure 4, Figure 5 and Figure 6. Evidently, the proposed method exhibits the best tracking performance. The performance indexes including the maximum absolute error (MAE) and the root mean square (RMS) values of the output tracking errors are presented in Figure 7 to further demonstrate the superiority of the proposed method.

### 4.3. Case II—Step Signal Tracking with Disturbance

In this case, the reference angular speed is set as square-wave signals with amplitudes of 10 °/s and 30 °/s separately. From time 1 s to 2 s, an external disturbance with an amplitude of 0.1 N·m is added to the ISIS system. The response curves are shown in Figure 8. Evidently, the proposed method exhibits the best disturbance rejection capability.

### 4.4. Case III—Stabilization with Disturbance

In the stabilization mode, the speed command is set as $\omega_{r}=0$. To simulate carrier vibration, a sinusoidal disturbance d=0.1sin(5πt) N·m is added to the system. The results are given in Figure 9. The figure exhibits that the proposed method exhibits the smallest speed and angular deviation. Figure 10 shows the amplitude frequency curve of Figure 9, and it indicates that the proposed method has the best disturbance rejection effect on the 2.5 Hz disturbance. The performance indexes, including the MAE and RMS of both speed and angular deviation, are shown in Figure 11, respectively, which further verifies the effectiveness of the proposed method.

## 5. Experiments

Experiments under operating conditions similar to those of the simulations were conducted to verify the effectiveness of the proposed method. The experimental setup of the ISIS is shown in Figure 12.

The ISIS is driven by DSP TM320F28335 with a sampling frequency of 1 kHz. The calculation of the proposed method includes three fundamental operations, integral, difference, and exponential operations. The calculation obtained using the proposed method is not substantially higher than that obtained using the traditional control method. All operations can be completed based on the hardware resource of the DSP; the complexity of the proposed method is acceptable, and the angular speed is measured using a gyro mounted on the camera, and an encoder is employed to measure the relative angle. Further, a visible camera is utilized to transmit a real-time image. Simultaneously, real-time data are transmitted to a computer via RS-422 serial port. The parameters of the ISIS system are $J_{n}$ = 0.000265 kg·m^2^ and $B_{n}$ = 0.00530 N·m·s. Meanwhile, the PID controller and the TSM controller with the traditional exponential reaching law is implemented. The parameters of different control methods are listed in Table 3.

### 5.1. Case I—Sinusoidal Signal Tracking

For this case, the reference angular speed is set as 30sin(2πt) °/s. The experimental response curves are shown in Figure 13. Since the ISIS system is evidently affected by the multiple disturbances, the proposed method exhibits the best tracking performance. Performance indexes including the MAE and RMS values of the output tracking errors are presented in Figure 14, which further verifies the effectiveness of the proposed method.

### 5.2. Case II—Step Signal Tracking with Disturbance

Here, the reference angular speed is set as square-wave signals with an amplitude of 30 °/s. From time 1 s to 2 s, a step voltage disturbance with an amplitude of 0.1 V is added to the motor voltage via program codes. The experimental response curves are shown in Figure 15. Evidently, the proposed method exhibits the best anti-disturbance ability.

### 5.3. Case III—Stabilization with Disturbance

The reference speed is fixed to zero in the stabilization mode. Sinusoidal disturbance with an amplitude of 0.1 V and frequency of 2.5 Hz is added to the motor voltage via program codes to simulate carrier vibration. Figure 16 shows that the proposed method has the smallest speed and angular deviation. The amplitude frequency curve of Figure 16 is given in Figure 17. The figure clearly shows that the proposed method possesses the strongest robustness against the external disturbance of 2.5 Hz. The performance indexes, including the MAE and RMS values of both the speed and angular deviation, are shown in Figure 18, respectively, which shows that the proposed method has the lowest speed and angular deviation, and smallest RMS.

The disturbance rejection effect of the simulation in Figure 10 differs from the experiment results in Figure 17. This is because solely mass unbalance torque and friction torque are considered in the simulation. However, in the actual system, there are other disturbances, such as cable restraint torque and sensors measurement error, apart from these two types. Both simulation and experiment results show the effectiveness of the proposed method in rejecting the main disturbance at 2.5 Hz.

## 6. Conclusions

In this study, the TSMC with NERL based on HOTSMO was investigated in the ISIS system. The TSMC based on the NERL method adjusts the switching gain as the sliding surface changes, owing to which the convergence speed of the TSMC is accelerated. An HOTSMO is further implemented to estimate the lumped disturbance of the ISIS system. Both the simulation and experimental results show that the proposed method exhibits the best tracking performance and the strongest disturbance rejection ability than PID and the traditional TSMC methods.

In the future, we plan to achieve a satisfactory performance of the tracking loop based on this study. The tracking loop generates the reference angular speed for the inner loop though visual signals, and the optical equipment faces the problem of time lapse during the identification of the target. Hence, we will aim to solve the time delay of the tracking loop.

## Figures and Tables

**Figure 1 sensors-20-03107-f001:**
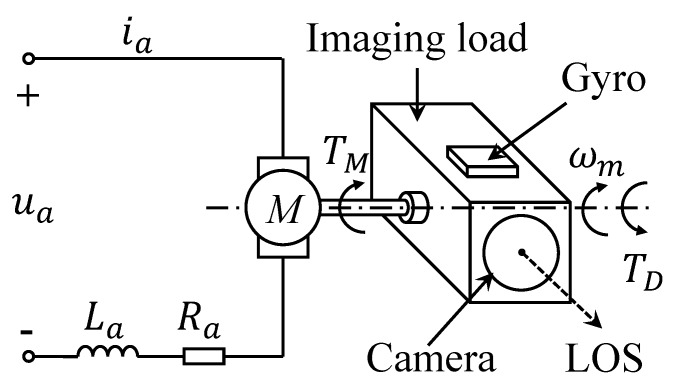
Schematic diagram of an inertial stabilization imaging sensor (ISIS) system.

**Figure 2 sensors-20-03107-f002:**
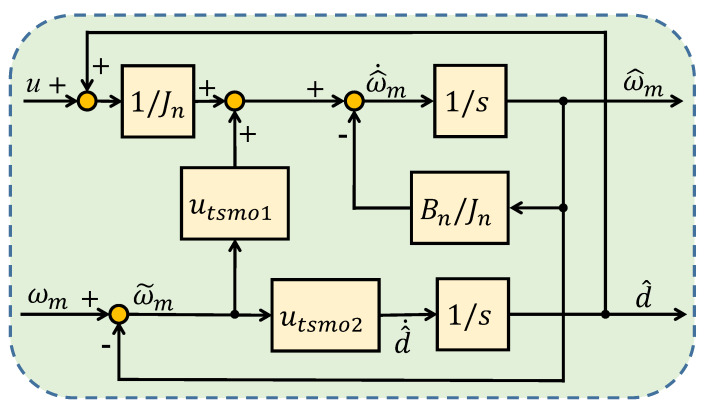
Block diagram of a high-order terminal sliding-mode observer.

**Figure 3 sensors-20-03107-f003:**
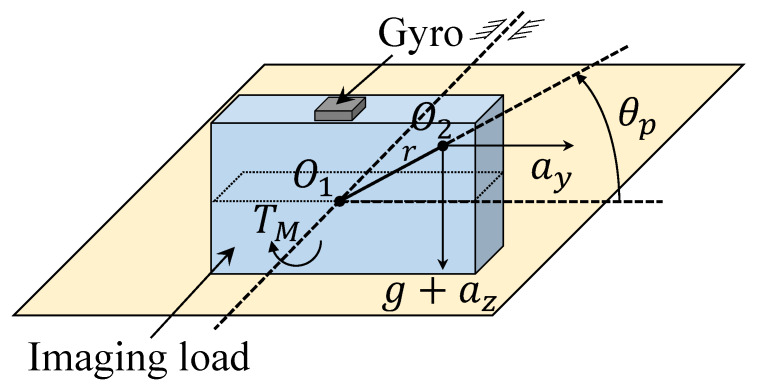
Schematic diagram of mass imbalance.

**Figure 4 sensors-20-03107-f004:**
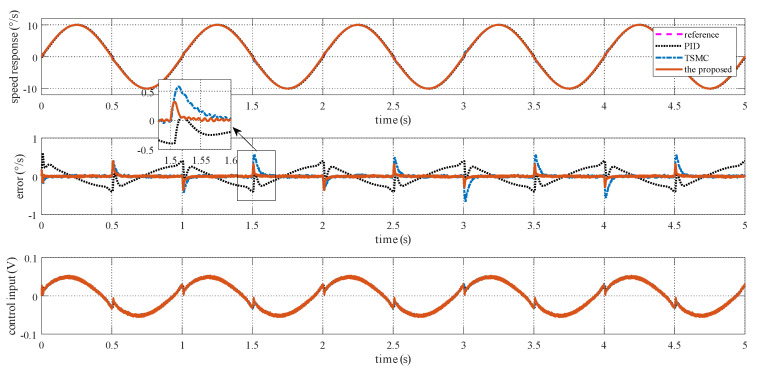
Simulation response curves to a sinusoidal reference signal with 10 °/s amplitude and 1 Hz frequency.

**Figure 5 sensors-20-03107-f005:**
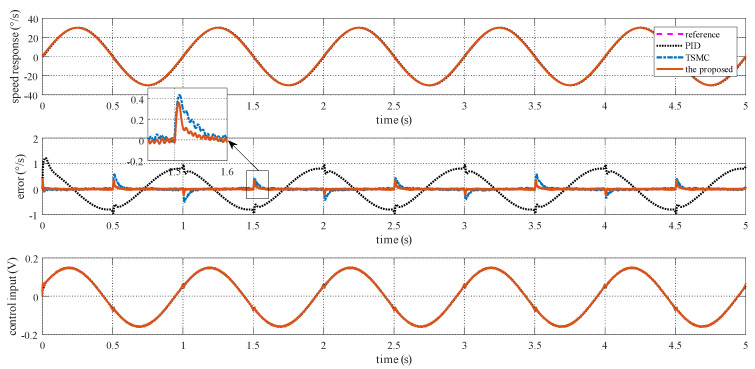
Simulation response curves to a sinusoidal reference signal with 30 °/s amplitude and 1 Hz frequency.

**Figure 6 sensors-20-03107-f006:**
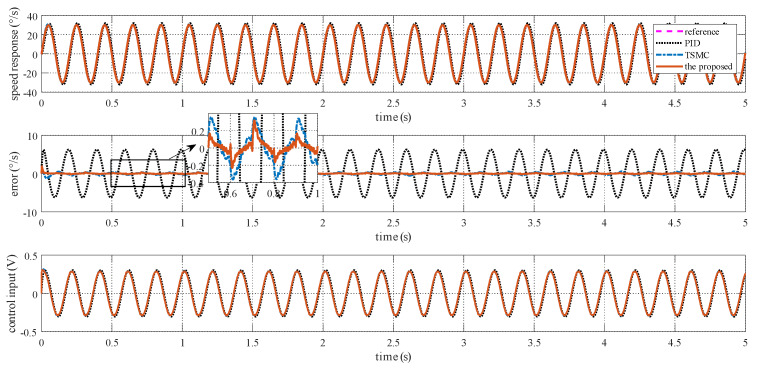
Simulation response curves to a sinusoidal reference signal with 30 °/s amplitude and 5 Hz frequency.

**Figure 7 sensors-20-03107-f007:**
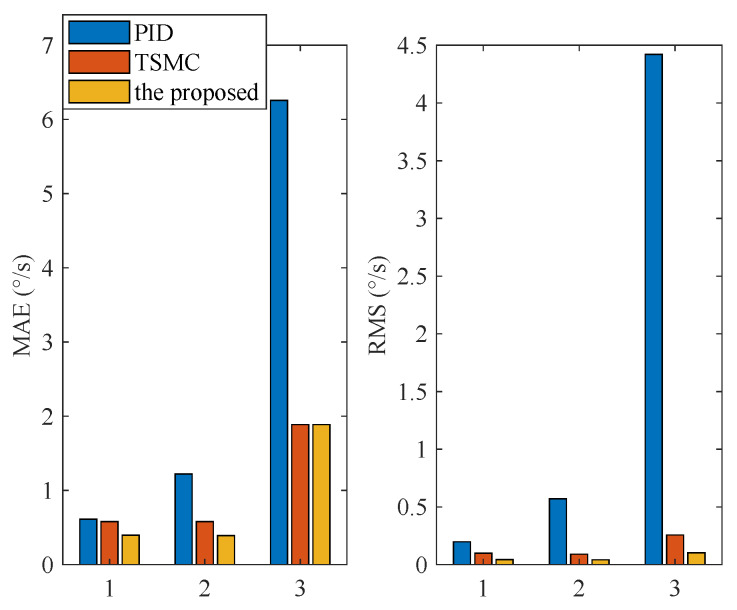
Simulation performance indexes under Case I with different tests.

**Figure 8 sensors-20-03107-f008:**
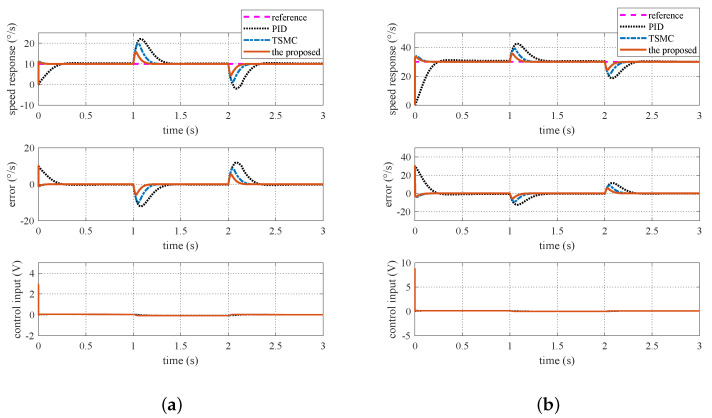
Simulation response curves under Case II: (**a**) Step reference signal with 10 °/s; (**b**) step reference signal with 30 °/s.

**Figure 9 sensors-20-03107-f009:**
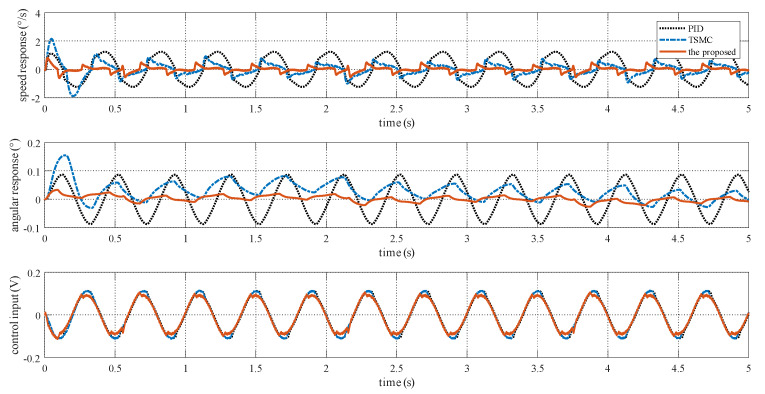
Simulation response curves under Case III.

**Figure 10 sensors-20-03107-f010:**
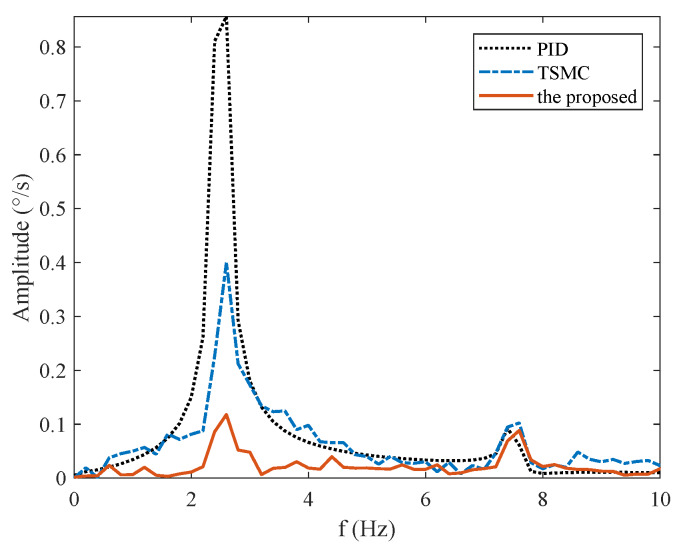
Frequency spectrum under Case III.

**Figure 11 sensors-20-03107-f011:**
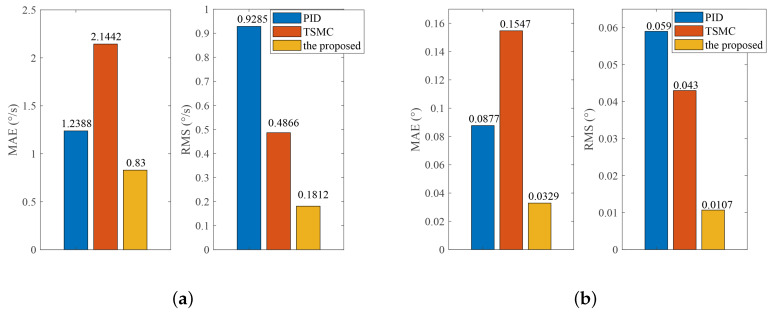
Simulation performance indexes under Case III: (**a**) Indexes of speed deviation; (**b**) indexes of angular deviation.

**Figure 12 sensors-20-03107-f012:**
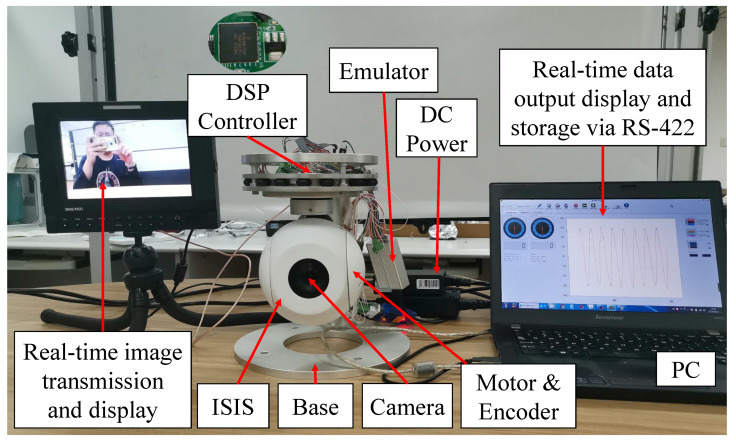
Experimental setup.

**Figure 13 sensors-20-03107-f013:**
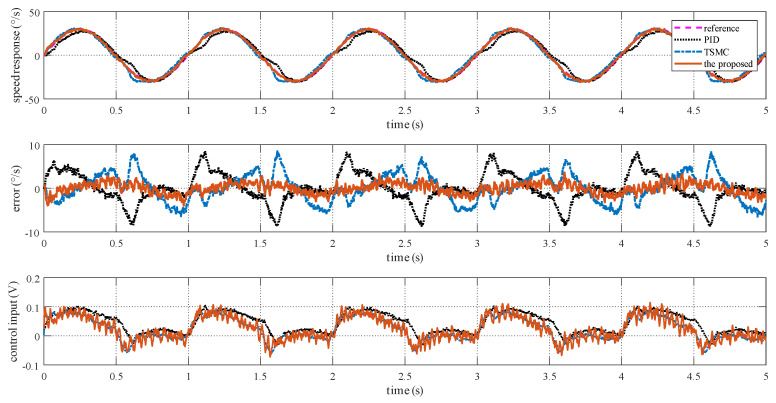
Experimental response curves under Case I.

**Figure 14 sensors-20-03107-f014:**
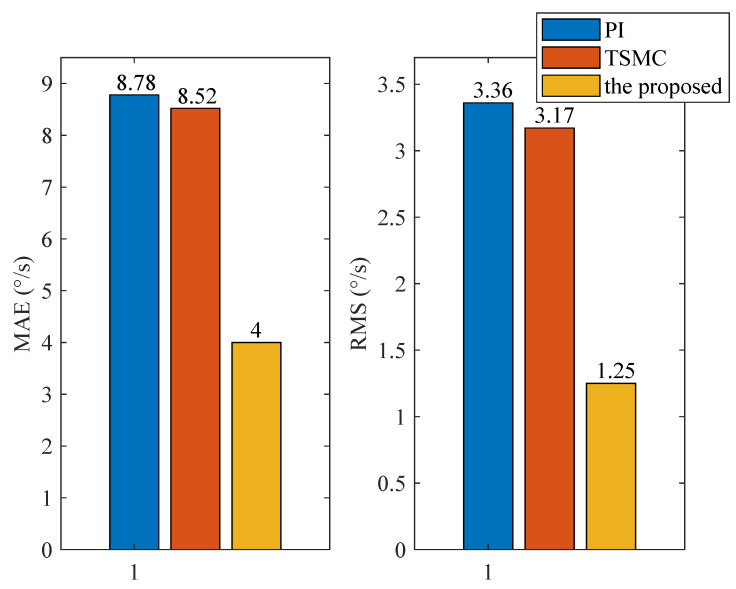
Experimental performance indexes under Case I.

**Figure 15 sensors-20-03107-f015:**
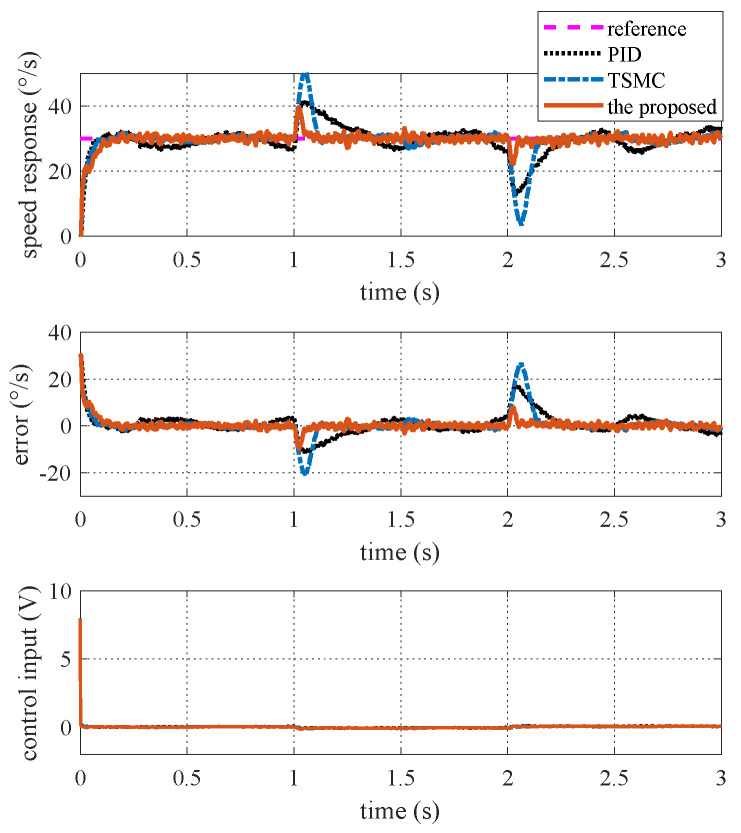
Experimental response curves under Case II.

**Figure 16 sensors-20-03107-f016:**
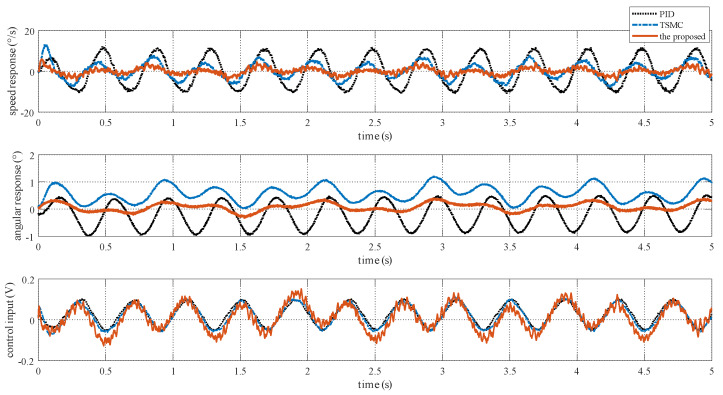
Experimental response curves under Case III.

**Figure 17 sensors-20-03107-f017:**
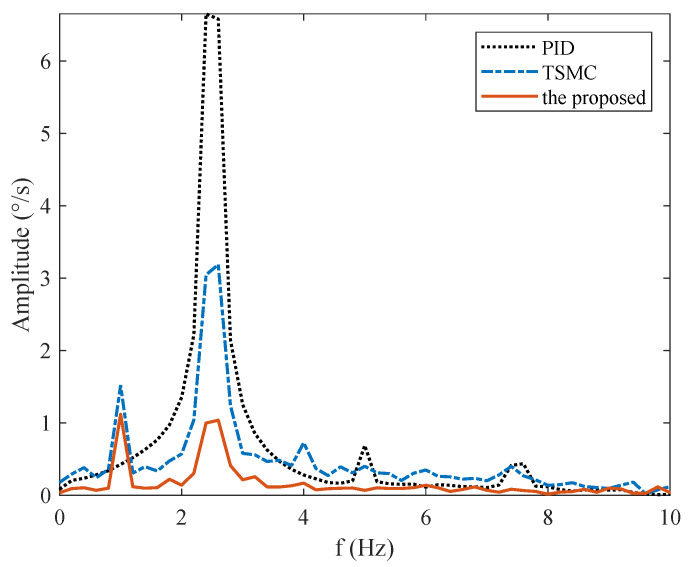
Frequency spectrum under Case III.

**Figure 18 sensors-20-03107-f018:**
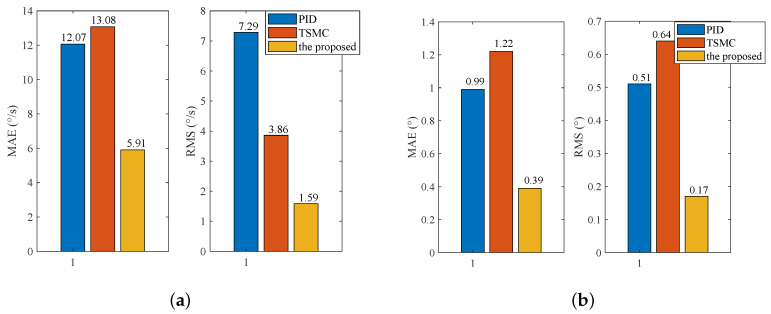
Experimental performance indexes under Case III: (**a**) Indexes of speed deviation; (**b**) indexes of angular deviation.

**Table 1 sensors-20-03107-t001:** Simulation parameters of the ISIS system.

Parameters	Description	Value
*J*	Moment of inertia	0.000265 kg·m^2^
*B*	Frictional coefficient	0.00530 N·m·s
$J_{n}$	Nominal value of *J*	1.1 J
$B_{n}$	Nominal value of *B*	1.1 B
*m*	Mass	1.5 kg
*r*	Centroid offset distance	5 mm
$a_{y}$	Horizontal acceleration	0.1 g
$a_{z}$	Vertical acceleration	0.1 g
$T_{c}$	Coulomb friction torque	0.001 N·m
$T_{s}$	Maximum static friction coefficient	0.01 N·m
σ	Viscous friction coefficient	0.0005 N·m/ °/s
$\omega_{s}$	Stribeck velocity	2 °/s

**Table 2 sensors-20-03107-t002:** Simulation parameters of different control methods.

Controller	Parameters
TSMC	$p=7,q=5,\alpha =0.01,k_{1}=6000,k_{2}=10$
the proposed	$p=7,q=5,\alpha =0.01,k_{1}=4000,k_{2}=5,$ δ=0.5, μ=10,β=0.05,m=5,n=3, $l_{1}=0.25,$ $l_{2}=1,T_{\omega }=100$

**Table 3 sensors-20-03107-t003:** Experimental parameters of different control methods.

Controller	Parameters
TSMC	$p=7,q=5,\alpha =0.005,k_{1}=14,000,k_{2}=5$
the proposed	$p=7,q=5,\alpha =0.005,k_{1}=8000,k_{2}=5,\delta =0.5,\mu =10,\beta =0.05,m=5,n=3,l_{1}=0.5,l_{2}=2,T_{\omega }=100$

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
