# Peer review of "Terminal Sliding Mode Control with a Novel Reaching Law and Sliding Mode Disturbance Observer for Inertial Stabilization Imaging Sensor"

_sensors, 2020, doi:10.3390/s20113107_

Round 1
Reviewer 1 Report
A terminal sliding mode controller (TSMC) based on a novel exponential reaching law (NERL) method with a high-order terminal sliding mode observer (HOTSMO) is suggested to overcome the complex nonlinearities of the control of inertial stabilization imaging sensors (ISISs). Comparative simulation and experimental results show that the proposed method achieves better tracking performance and stronger robustness than traditional TSMC. We recommend publishing after minor repairs.
Some comments and suggestions are as follow:
- In Equation (20), what r means is not mentioned in the context.
- In line 158, “\alpha > 0” => “\beta >0”.
- From Figure 10 and Figure 17 in the simulation and experiment, there are differences in the disturbance rejection effect of case â…¢. How to explain these differences?
Reviewer 2 Report
The paper deals with a novel technique to design a so-called terminal sliding mode control (TSMC) using a new exponential reaching law method, and a high-order terminal sliding observer (HOTSMO) to perform disturbance rejection control for an inertial stabilization imaging sensor (ISIS).
First, would like to mention that I am not an expert on this particular physical system. Analyzing the model of the system which is pretty much like a regular DC motor, I cannot help wondering why such a sophisticated control technique like sliding mode is necessary? The arguments in the introduction are not quite convincing to the untrained eye.
In detail, the system is linear, of low dimensions in both the number of states and number of inputs and outputs. I wonder why a simple PID control is not enough for the proposed goals? One reason that springs to mind is the lack of robustness. The authors should comments on this issue to make the manuscript more self-contained.
Speaking of robustness and in particular of the desired goal to reject a specific class of disturbances, why not apply simple loopshaping (not necessarily H-infinity control) to achieve robust stability as well as robust performance? In the technique proposed in the paper the closed-loop system becomes nonlinear, of course, due to the sliding manifold and the authors apply a Lyapunov stability analysis. Concerning the Lypaunov stability analysis, I am not totally convinced that the derivative of V is negative definite, as in (17) or (29). Isn't the application of the LaSalle principle more suitable here?
At the end of the day, I think the paper could benefit from being more self-contained from a control system theoretic perspective. A comparison to classic control is called for.
The paper is nicely written and the English is good. However, slight mistakes are here and there and the authors ought to check them and correct them.
Once these revisions have been completed, the paper may proceed to publication.
Reviewer 3 Report
This paper addresses the Sliding Mode Control problem for Inertial Stabilization Imaging Sensor. The paper is well written and the topic is very interesting. Both numerical and experimental results are presented. Follow some comments from the reviewer:
The number of acronyms should be reduced, that would improve the readability of the paper.
The authors talk about parameter uncertainties, but it seems that this concept has not been employed. The authors should include some discussion about existing techniques to deal with parameter uncertainties, inexactly scheduling parameters for instance:
[A] On inexact LPV control design of continuous-time polytopic systems, IEEE Transactions on Automatic Control, vol. 53, no. 7, pp. 1674-1678, Aug. 2008. 10.1109/TAC.2008.928119
[B] A new approach to handle additive and multiplicative uncertainties in the measurement for LPV filtering, International Journal of Systems Science, 47:5, 1042-1053, DOI: 10.1080/00207721.2014.911389
The authors should further discuss Assumption 1. When this requirement is not met what can be done?
Theorem 1 should be rewritten. The conditions should be given in the Theorem not in the Proof. The Proof. should be used to certify that, if the conditions proposed in the Theorem hold the system converges in finite time. At this point, it is not clear what (17) is.
It is not clear if Theorem 2 assumes Assumption 1 holds.
In case I, it is hard to see any improvement in the speed response and in the control input.
In case II which method uses less control energy?
In Table 3 it can be seen that the proposed method employs more parameters than the TSMC, what is the impact of the choice of the parameters in the effectiveness of the control? The authors should give some guidance to the reader to choose the parameters.
In Figure 13 the control input provided by the proposed method is chattering, how to remove this effect? The authors should include an index associated with the control input also, not only the error.
Page 13: "The parameters of the ISIS system is"
Round 2
Reviewer 3 Report
The paper has been revised satisfactorily.
I have no further comments.